# Retrospective Self-Reports of How Adolescent Substance Use Changed with the COVID-19 Pandemic

**DOI:** 10.3390/ijerph192013680

**Published:** 2022-10-21

**Authors:** Janni Leung, Catherine Quinn, Molly Carlyle, Rhiannon Ellem, Calvert Tisdale, Lily Davidson, Melanie J. White, David J. Kavanagh, Leanne Hides

**Affiliations:** 1Lives Lived Well Group (LLW), School of Psychology, The University of Queensland, St. Lucia, QLD 4072, Australia; 2National Centre for Youth Substance Use Research (NCYSUR), The University of Queensland, St. Lucia, QLD 4072, Australia; 3School of Psychology and Counselling, Queensland University of Technology, Brisbane City, QLD 4000, Australia

**Keywords:** alcohol, underage drinking, cannabis, marijuana, adolescent behavior, adolescent health, pandemics, COVID-19

## Abstract

The final year of high school is a challenging phase of adolescents’ lives and substance use can play an important role. We examined changes in the frequency and quantity of alcohol and cannabis use, and demographic correlates among Grade 12 students of 2020. Students (N = 844) from nine schools retrospectively self-reported changes in substance use after the easing of COVID-19 lockdowns (back to school), compared to before the pandemic. Changes in use were examined with age, gender, Aboriginal or Torres Islander, parental and family characteristics, and truancy. Thirty-one percent of students reported that they used alcohol less frequently, and 24% reported that they used it more frequently compared to pre-COVID-19. Most students (46%) reported that they used cannabis less, while a subset reported using more frequently (22%). A history of truancy was associated with an increased frequency (OR = 2.13 [1.18–3.83]) of cannabis use. A substantial minority of adolescents used more alcohol and cannabis after the initial COVID-19 lockdown period. Students in their final year who reported increased use may benefit from increased support to manage their substance use.

## 1. Introduction

The Coronavirus (COVID-19) pandemic is a global health crisis that led to vast disruptions to social contact [1], and likely disproportionately impacted young people who heavily rely upon peer relations for social support [2]. Studying from home arrangements restricted peer-to-peer social interaction opportunities, while also reducing access and exposure to substance use because of increased parental supervision [3]. Pre-existing risk factors associated with substance use (e.g., socio-demographic factors, parental and family-related factors, and missing school [4,5,6] may have exacerbated substance use due to disruptions caused by the pandemic.

Due to these concerns, the impacts of the pandemic on substance use in young people have received some investigation [7,8]. A population-based longitudinal study of 59,700 Icelandic adolescents reported reductions in cigarette, vaping, and alcohol use [9]. A retrospective study among Canadian adolescents similarly reported no change in absolute numbers of adolescents using alcohol, and reductions in the number of adolescents who were vaping, binge drinking, and using cannabis [10]. However, for those adolescents who were already using substances, the frequency of cannabis and alcohol use increased [10], which suggests that a subset of adolescents may be at risk of increased substance use during and after the pandemic.

Given the long-term harms of increased substance use on adolescent neurodevelopmental health [11], it is critical to more comprehensively address substance use through the pandemic [12]. Substance use issues may be particularly prominent in senior students in their final years of schooling because of increased stress with final exams and increased access and exposure to substances.

This study aimed to examine retrospective self-reports of how alcohol and cannabis use changed with the COVID-19 pandemic among adolescents who reported their use. We conducted quantitative analysis on cross-sectional surveys of adolescents in June-October 2020—after the initial lockdown restrictions were eased in Queensland, Australia. This initial lockdown included the closure of schools from March 2020, with stay-at-home learning replacing classrooms until restrictions were eased and classrooms were reopened in June 2020. We focused on grade 12s of 2020, who were in their final year of high school when the COVID-19 social distancing protocols were at the highest level, because this important year in their lives was severely impacted by the pandemic. We examined both changes in frequency and quantity of use in existing users. In addition, we examined what student characteristics factors were associated with changes in alcohol and cannabis use.

## 2. Materials and Methods

### 2.1. Study Design

This is a cross-sectional quantitative analysis of retrospective self-reports in school surveys.

### 2.2. Setting

Data were from The Adolescent Aware project, a Queensland University of Technology (QUT) and University of Queensland (UQ) collaboration. It is a school-based study, which has collected self-report survey data from cohorts of students at independent high schools in South-East Queensland, Australia each year between 2015 and 2020.

Researchers visited participating schools to administer the survey during Term 3–4 of 2015, when students were in Grade 7 and 8, and then re-surveyed students every year till they completed Grade 12 (i.e., in 2019 and 2020). Opt-out parental consent and active student self-consent was sought for students prior to each survey, which were completed either online, or via paper and pen administration, depending on the method most convenient to each school.

Participants of the current study included the Grade 12 students of 2020. Data were collected from nine high schools between 12 June to 12 October 2020. In 2020, schooling had been disrupted by the COVID-19 pandemic and the students experienced a period of studying from home from 30 March to 25 May 2020 (see Figure 1). Data were collected from participating schools between 12 June and 12 October 2020 (0.5–4.5 months after lockdown), during the initial months when students physically returned to school, after the stay-at-home period.

### 2.3. Informed Consent

All procedures followed were in accordance with the ethical standards of The University of Queensland Human Research Ethics Committees and with the Helsinki Declaration of 1975, as revised in 2000. Informed consent was obtained from all participants included in the study.

### 2.4. Variables

#### Changes in Alcohol and Cannabis Use

Changes in substance use compared to before the COVID-19 pandemic were measured using retrospective self-report items. Students were first asked if they had ever used alcohol and if they had ever used cannabis. Those who reported that they had consumed a full drink of alcohol in their lifetime were asked about their changes in alcohol consumption, and those who reported that they had used cannabis in their lifetime were asked about their changes in cannabis consumption.

Four items assessed changes in the frequency and quantity that students were using alcohol and cannabis after the lockdown phase of 2020. Students were asked to retrospectively compare how often (frequency) and how much (quantity) they were using alcohol and cannabis “NOW” (at the time of the survey), compared to before the pandemic and before social distancing restrictions were put in place (i.e., end of March 2020). Responses were categorised as “Less”, “No change”, or “More”.

### 2.5. Student Characteristics

Student characteristic variables examined were age, gender, Aboriginal or Torres Strait Islander status, parental country of birth, parental relationship status, number of family vehicles (proxy measure for socio-economic status), and truancy in the past year.

### 2.6. Statistical Analysis

Participants’ characteristics were described with 95% confidence intervals (CIs) reported. Among students who had ever used alcohol and cannabis, we presented proportions of students self-reporting less, no change, or more of their use compared to before the pandemic. We presented results on frequency and quantity of use separately.

The association between changes in substance use and characteristic variables were examined using ordinal logistic regressions. Four models were conducted on the outcomes of: (1) alcohol frequency; (2) alcohol quantity; (3) cannabis frequency; (4) cannabis quantity. The response categories were ordered from “Less” to “No change” to “More”, to represent increasing frequency or quantity consumed. Hence, “increased” represented increasing order which takes into account the three levels of the response categories. Odds ratios of increased alcohol and cannabis use were estimated by all the characteristic variables entered together in the models. The models also controlled for days to the end of the year to adjust for the different times when the data were collected. Levels of missing data were minimal at 1.3%, so missing data were excluded listwise for consistency across analyses. Analyses were conducted in SPSS 27. We ran four models, so we adjusted the family-wise error by using 0.0125 (0.05/4) as our *p*-value cut-off for significance.

## 3. Results

### 3.1. Participants and Characteristics

An initial sample of 1127 students were invited to participate, from whom 855 students responded and provided consent (76% response rate). A further 11 participants (1.3%) were excluded because of missing data on the response items (n = 7 missing for alcohol, n = 10 missing for cannabis [some of these had missing data for both]). A final sample of N = 844 participants was available.

As Table 1 shows, participants had an average age of 17 years with little variation. Slightly over half were male, and a small proportion identified as Aboriginal or Torres Strait Islander. The majority had both parents born in Australia, had parents living together, and owned two or more family vehicles. Slightly over 1 in 10 had a history of truancy. The 73.2% (n = 610) who had ever had a full drink of alcohol, and 27.9% (n = 232) who had ever tried cannabis were included in the analysis below on changes in consumption levels.

### 3.2. Changes in Alcohol and Cannabis Use

Among Grade 12s of 2020, there was variation in responses of changes in alcohol and cannabis use (see Figure 2). When asked to compare the frequency and quantity of alcohol use now with before the COVID-19 restrictions, the greatest proportion of students reported no change, a third reported less use, and a substantial minority reported that they used more frequently. For cannabis use, more students reported that they used less frequently and, in less quantities, but over 1 in 5 reported that they used more.

### 3.3. Associates of Increased Substance Use

Table 2 presents the results of the ordinal logistic regression analyses on self-reported increasing order of substance use compared to pre-COVID-19. Older students were more likely to report increased consumption in frequency (OR = 1.41 [0.96–2.05]) and quantity (OR = 1.39 [0.95–2.04]) for alcohol use, but these results did not reach statistical significance after adjusting for all the other socio-demographic characteristics. Having parents who were born in Australia was associated with higher odds of reporting increased alcohol use, specifically a significant association was observed between having a parent born in Australia and increased quantity of alcohol used (OR = 2.11 [1.26–3.51]).

For cannabis use, a history of truancy was associated with increased frequency (2.13 [1.18–3.83]) and quantity of cannabis use (2.10 [1.17–3.78]). However, only the result for increased frequency (*p* = 0.0117) was statistically significant, and the result for increased quantity (*p* = 0.0133) did not reach statistical significance after adjusting for family-wise error. Students with those parents born in Australia were less likely to have increased cannabis use, while those with a lower number of vehicles in the family were more likely to increase their use, but these were not statistically significant in the adjusted models.

## 4. Discussion

In this cross-sectional study of final-year school students, we examined retrospective self-reports of changes in the frequency and quantity alcohol and cannabis use after the initial easing of the lockdowns, compared with the period before the COVID-19 pandemic in students who had ever used these substances. We focused on the period just after COVID-19 social distancing protocols were at the highest level in our setting. Almost half of the students reported drinking the same amount of alcohol, but among the remaining half, there was an almost equal split of students reporting that they used less or more in frequency and quantity. Among students who had used cannabis previously, cannabis use was unchanged in around a third, around twice as many reported that they used less, but still a substantial minority reported that they used more. Increased use of cannabis is a risk factor because longitudinal studies have found that one in three adolescents who used cannabis frequently developed cannabis use disorders later in life [13].

The current findings build upon existing retrospective research, demonstrating that a subset of adolescents who had used alcohol or cannabis in their lifetime may have used more during or after the pandemic [10]. We add to this research by examining quantity in addition to frequency, because among adolescents who many were not drinking in high frequencies regularly, high quantity of alcohol or cannabis use that result in a high level of intoxication puts the adolescents at risk of adverse consequences.

Our findings are in line with existing research that reported reductions in cannabis use in a population-based adolescent study [9]. The results may not apply to other settings, because there are jurisdictional differences in cultural and political approaches to alcohol access throughout the pandemic. Policies targeting alcohol use within Australia have been comparatively less restrictive to other countries over the pandemic period [14]; For example, an increased use of home delivery of alcohol made it more difficult to enforce minimum age limits [15].

Other social components may include parental drinking, which has already been identified as a risk factor for greater adolescent alcohol use during the pandemic. We found that students who reported having a parent who were born in Australia were more likely to self-report increased alcohol use compared to pre-COVID-19. There may be cultural differences in family relations related to alcohol use that warrant further investigation. Over the COVID-19 pandemic, alcohol advertisers increasingly used social media to encourage adults to drink at home [16]. This raises the question of whether increased parental drinking at home impacted adolescent drinking. The current study did not assess the context of their drinking habits, including where and with whom they drank. A recent longitudinal adolescent study in US found that before COVID-19, almost all parents forbade adolescent drinking with the family, but data collected during the lockdown showed that 1 in 6 allowed it [17], with parents who drank heavily being more lenient in allowing adolescent drinking during lockdown. Interventions to support parental alcohol use could have secondary benefits to children’s health, and family-based interventions could be an effective treatment option for addressing substance use in both parents and adolescents [18,19].

Older students in our study were not significantly more likely to report drinking or using cannabis more after the onset of the COVID-19 pandemic. These results imply that instead of older age, other socio-economic factors may be stronger determinants of alcohol use behaviors in adolescents. Previous research on the initiation of substance use have shown increased use related to developmental changes in alcohol consumption [20]. In our context, most adolescents were aged 17, and approaching the legal drinking age of 18 in Australia.

Truancy was associated with increased self-reports of cannabis use. Previous research has shown that adolescent cannabis use is associated with school drop-out, which may position them onto a disadvantaged pathway for the rest of their lives [11]. While our results cannot show the direction of influence, our results imply that the minority of students who increase their cannabis use may benefit from treatment and support services to manage their use and prevent cannabis-related negative outcomes. Online or e-health options may be an alternative for youth who may not wish to access face-to-face help. Randomized controlled trials have shown that internet-based preventive interventions are effective for reducing alcohol and cannabis use, as well as decreasing truancy and improving mental health outcomes in school students [21].

### Limitations

The retrospective nature of the items used to measure changes in substance use represents a significant limitation of this study. The items assessing the impact of the COVID-19 lockdown on the 2020 Grade 12 students’ substance use required students to self-quantify a change compared to “now”, which may result in differing interpretations of “more”, “same” and “less”. This item has its limitations because of the lack of quantification of “more” or “less” use. Future research would benefit from examining the extent of change in more detail. For example, did students who increased their frequency of use go from using once a month to once a week, once a day, or did they go from none to once a month. This would be especially important for accurately understanding risk levels because the risk of substance use highly depends on the actual frequency and quantity used.

Additionally, survey administration during 2020 spanned a period of 4 months (June–October). Thus, “now” at the time of the survey, differed across schools in relation to the peak of the COVID-19 lockdown, and may be less reliable for students surveyed towards the latter end of the year due to greater recall bias. Students who completed the survey later in the year would also have been older and therefore likely to consume more than younger students. To mitigate some of these limitations, our primary analyses on the extent of changes in consumption adjusted for age and days to the end of the year.

Further, students were from non-governmental independent high schools and may not be representative of the wider senior student community. Independent schools form a small part (11.4%) of the overall Australian school system and they typically enroll students from higher socio-economic backgrounds [22]. Findings may not be applied to other settings, because there were variations in lockdown policies across Australia, with lockdowns in Queensland in general being less stringent than those of other states. Students in settings where the COVID-19 pandemic may have hit harder may be more impacted than those in our study. Our study may also have underestimated both the extent of cannabis use and its relationship with truancy, if some students who were using more cannabis were not present at school to participate in our study.

Unfortunately, we do not have information on the initiation of use of alcohol and cannabis during the pandemic. If we had such data, we would need comparable data from a previous cohort of same age adolescents from previous years before the pandemic, because initiation of these substances may have occurred as adolescents aged, so it is difficult to attribute the initiation of use to the pandemic. Future studies on differences in substance use across cohorts of adolescents who have lived through the COVID-19 pandemic, compared to those who had not, may provide some indication of possible impacts. We have revised our discussion section to add this important point.

Our study was limited by the availability of variables collected in the survey. For example, number of family vehicles was our only indicator for the socio-economic situation of the students. Information on this variable was collected using the options of “no”, “yes, one car”, “yes, two or more cars”. In our sample, there was a very small proportion of students who reported that they had no family vehicles, so they were combined into the group of students who had one family vehicle. There was a high proportion of students who reported two or more family vehicles, but we do not know the composition within that group. Having three or more vehicles may be an indicator of higher financial status, but we were not able to separate those families that had three cars from those with two, so future research that plans to collect this information could consider providing response options beyond two cars. In addition, we did not have information on other variables that may be particularly important during the COVID-19 pandemic, such as frequency of social contact, loneliness, and family strain [7,23,24], which may affect substance use. Future research that collects more detailed data could provide important information for understanding the psychological and social context underlying substance use in adolescents.

Future research that compares students from different countries, with different experiences of the COVID-19 pandemic and consequent lockdowns, will extend upon this research. Specifically, this research could compare substance use outcomes of high school students across different settings, as well as same-age adolescents outside of school settings who may be at increased risks of substance use.

## 5. Conclusions

Based on retrospective reports, most of this sample of Grade 12 students either used alcohol and cannabis less or at the same levels as before COVID-19, but a minority reported that they used more alcohol or cannabis. Students in their final year who reported increased use may benefit from increased support to manage their substance use, both for current functioning and functioning in later life.

## Figures and Tables

**Figure 1 ijerph-19-13680-f001:**
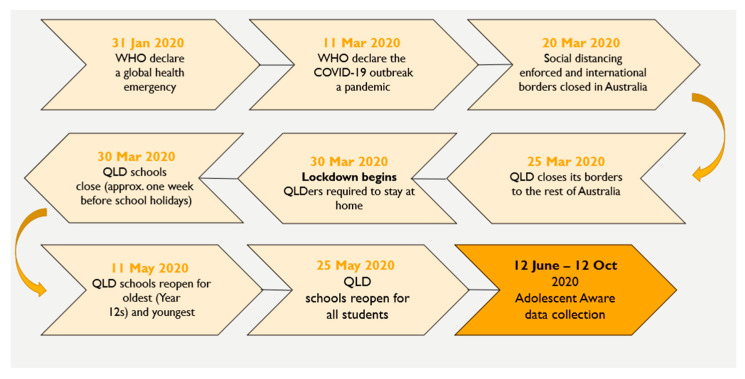
The 2020 Adolescent Aware data collection period in relation to school closure and social distancing policies of the 2020 COVID-19 pandemic in Queensland (QLD) Australia.

**Figure 2 ijerph-19-13680-f002:**
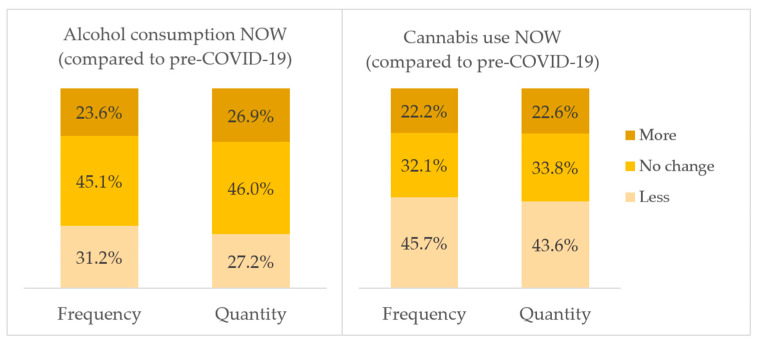
Self-reported changes in alcohol and cannabis use in relation to the COVID-19 pandemic in 2020 among Grade 12 students.

**Table 1 ijerph-19-13680-t001:** Characteristics of Grade 12s of 2020 (N *=* 833; mean age *=* 17.2 [17.1–17.2]).

	N	% [95% CI]
Gender		
Male	502	60.3 [56.8–63.6]
Female	331	39.7 [36.4–43.2]
Aboriginal or Torres Strait Islander		
No	803	96.4 [95.1–97.6]
Yes	30	3.6 [2.4–4.9]
Parental country of birth		
Both born in Australia	506	60.7 [57.4–64.2]
One parent born in Australia	163	19.6 [16.8–22.3]
Neither born in Australia	164	19.7 [16.9–22.1]
Parental relationship status		
Living together	637	76.5 [73.5–79.4]
Not living together	196	23.5 [20.6–26.5]
Family vehicles		
None or one	82	9.8 [7.7–12.0]
Two or more	751	90.2 [88.0–92.3]
Truancy in past year		
No	727	87.3 [85.0–89.3]
Yes	106	12.7 [10.7–15.0]
Ever drank alcohol		
No	223	26.8 [23.8–30.0]
Yes	610	73.2 [70.0–76.2]
Ever used cannabis		
No	601	72.1 [68.9–75.3]
Yes	232	27.9 [24.7–31.1]

**Table 2 ijerph-19-13680-t002:** Ordinal logistic regression on increasing order of alcohol and cannabis use compared to pre-COVID-19.

	Increased Consumption Compared to Pre-COVID-19			
	OR (95% CI)	*p*	OR (95% CI)	*p*
	*Alcohol*			
	*Frequency*		*Quantity*	
Age	1.41 [0.96–2.05]	0.078	1.39 [0.95–2.04]	0.091
Gender				
Male	0.71 [0.46–1.09]	0.120	0.70 [0.45–1.08]	0.104
Female	1.00 [ref]		1.00 [ref]	
Aboriginal or Torres Strait Islander				
No	1.04 [0.47–2.27]	0.930	0.91 [0.42–1.93]	0.797
Yes	1.00 [ref]		1.00 [ref]	
Parental country of birth				
Both born in Australia	1.33 [0.87–2.02]	0.184	1.62 [1.06–2.46]	0.025
One parent born in Australia	1.54 [0.93–2.55]	0.096	**2.11 [** **1.** **26–3.51** **]**	**0.** **004**
Neither born in Australia	1.00 [ref]		1.00 [ref]	
Parental relationship status				
Living together	1.00 [0.69–1.44]	0.979	0.95 [0.66–1.38]	0.797
Not living together	1.00 [ref]		1.00 [ref]	
Family vehicles				
None or one	1.34 [0.77–2.32]	0.302	1.46 [0.84–2.55]	0.183
Two or more	1.00 [ref]		1.00 [ref]	
Truancy				
Yes	1.26 [0.81–1.97]	0.310	1.23 [0.80–1.88]	0.355
No	1.00 [ref]		1.00 [ref]	
	** *Cannabis* **			
	** *Frequency* **		** *Quantity* **	
Age	0.69 [0.35–1.36]	0.284	0.66 [0.33–1.31]	0.237
Gender				
Male	1.29 [0.59–2.81]	0.525	1.67 [0.75–3.71]	0.205
Female	1.00 [ref]		1.00 [ref]	
Aboriginal or Torres Strait Islander				
No	1.28 [0.36–4.58]	0.702	1.11 [0.33–3.81]	0.864
Yes	1.00 [ref]		1.00 [ref]	
Parental country of birth				
Both born in Australia	0.59 [0.30–1.16]	0.125	0.60 [0.31–1.17]	0.134
One parent born in Australia	0.80 [0.35–1.80]	0.583	1.08 [0.47–2.47]	0.852
Neither born in Australia	1.00 [ref]		1.00 [ref]	
Parental relationship status				
Living together	0.89 [0.50–1.57]	0.684	0.89 [0.50–1.59]	0.701
Not living together	1.00 [ref]		1.00 [ref]	
Family vehicles				
None or one	2.46 [0.99–6.10]	0.052	2.12 [0.85–5.25]	0.107
Two or more	1.00 [ref]		1.00 [ref]	
Truancy				
Yes	**2.13 [1.18–3.83]**	**0.012**	2.10 [1.17–3.78]	0.013
No	1.00 [ref]		1.00 [ref]	

Note. The response categories were ordered from “less” to “no change” to “more”, to represent increased frequency or quantity consumed. All models adjusted for age, gender, Aboriginal or Torres Strait Islander, parental country of birth, parental relationship status, family vehicles, truancy, school, and days to the end of the year. A *p*-value cut-off of 0.0125 (0.05/4) was used to infer significance because four models were conducted.

## Data Availability

Not applicable.

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
