# Peer review of "Retrospective Self-Reports of How Adolescent Substance Use Changed with the COVID-19 Pandemic"

_ijerph, 2022, doi:10.3390/ijerph192013680_

Round 1

Reviewer 1 Report

The authors identified an important problem of changes in adolescent substance use during the pandemic and the large sample size of the participant pool is excellent. However, given the set-up of the introduction, I expected the authors to test different predictors in the models – such as frequency of social contact, loneliness, family strain, instead of demographic variables. The introduction should set up the variables that are represented in their model. 

Methods:

The authors conduct 4 tests of the same model but do not include an adjustment of their p-value for the family-wise error. If a full family-wise error correction was applied (.05/4 = .0125), then most of their results would not be significant and only variables such as speaking English at home and having native parents would be associated with differences in substance use. This limits the paper to largely null findings. 

Results:

I would consider dropping the variable of the language spoken at home from your analyses as it seems likely that you would have a very small number of cases in each group of the analytic model, given the low base rate of another language being spoken at home alone. Did you test for multicollinearity between your variables?

The results of truancy with cannabis use are interpreted as increased with a history of truancy in the text but the table has the reverse. Which is correct?

Discussion:

Be consistent about COVID vs COVID-19 throughout the paper.

The results of the paper show marginal significance and the authors need to make a cogent argument for why their specific independent variables were included and why these results should matter for addressing adolescent substance use. If possible, if the authors have access to alternative predictors, then including them would significantly improve the importance and implications of the manuscript. 

Reviewer 2 Report

When you mention early in the introduction that COVID reduced access to and exposure to substance use and increased parental supervision, please provide a citation if available.

Please provide a citation for the statement in the introduction beginning, “Given the long-term harms of increased substance use on adolescent neurodevelopmental health…”

Was there an assessment of initiation of use of alcohol and cannabis? It would be interesting to see how many students reported first using alcohol or cannabis during the pandemic, particularly because of the previous studies that haven’t reported many changes in the numbers of adolescents who use substances, but rather changes in the amount or frequency of use.

Is there a reason that the family vehicles variable was not treated as a continuous variable? It seems like most families had 2 or more vehicles. I’d imagine it could at least be worth separating those families that had three cars from those with two, as that might be an indicator that mom, dad, and child have their own cars. The presence of a third car might be important for understanding independence and possibly greater access to substances.

It's worth pointing out in the limitations section that an additional drawback is related to the lack of quantification of “more” or “less” use. You could mention that future research would benefit from examining the extent of change in more detail—i.e., did students who increased their frequency of use go from using once a month to once a week, once a day, etc. This would be especially important for accurately understanding risk levels.
